# Long non-coding RNA SNHG17 may function as a competitive endogenous RNA in diffuse large B-cell lymphoma progression by sponging miR-34a-5p

Shengjuan Lu[1‡], Lin Zeng[1‡], Guojun Mo[2,3‡], Danqing Lei[3], Yuanhong Li[2], Guodi Ou[4], Hailian Wu[3], Jie Sun[1], Chao Rong[1], Sha He[1], Dani Zhong[1], Qing Ke[1], Qingmei Zhang[5,6], Xiaohong Tan[1], Hong Cen[1], Xiaoxun Xie[5,6]*, Chengcheng Liao[1]*

1 Department of Hematology/Oncology, Guangxi Medical University Cancer Hospital, Nanning, China,
2 Department of Pharmacy, The Second Affiliated Hospital of Guangxi Medical University, Nanning, China,
3 Life Sciences Institute, Guangxi Medical University, Nanning, China, 4 Pharmaceutical College, Guangxi Medical University, Nanning, China, 5 Department of Histology and Embryology, School of Pre-clinical Medicine, Guangxi Medical University, Nanning, China, 6 Key Laboratory of Early Prevention and Treatment of Regional High Frequency Tumor (Guangxi Medical University), Ministry of Education, Nanning, China

‡ SL, LZ, and GM are equal first authors on this work.
* xxiaoxun@163.com (XX); liaochengcheng@gxmu.edu.cn(CL)

**Data Availability Statement:** All relevant data are within the paper and its Supporting Information files.

## Abstract

We investigated the functional mechanism of long non-coding small nucleolar host gene 17 (SNHG17) in diffuse large B-cell lymphoma (DLBCL). lncRNAs related to the prognosis of patients with DLBCL were screened to analyze long non-coding small nucleolar host gene 17 (SNHG17) expression in DLBCL and normal tissues, and a nomogram established for predicting DLBCL prognosis. SNHG17 expression in B-cell lymphoma cells was detected using qPCR. The effects of SNHG17 with/without doxorubicin on the proliferation and apoptosis of DoHH2 and Daudi were detected. The effects of combined SNHG17 and doxorubicin were analyzed. The regulatory function of SNHG17 in DLBCL was investigated using a mouse tumor xenotransplantation model. RNA sequencing was used to analyze the signaling pathways involved in SNHG17 knockdown in B-cell lymphoma cell lines. The target relationships among SNHG17, microRNA, and downstream mRNA biomolecules were detected. A higher SNHG17 level predicted a lower survival rate. SNHG17 was highly expressed in DLBCL patient tissues and cell lines. We established a prognostic model containing SNHG17 expression, which could effectively predict the overall survival rate of DLBCL patients. SNHG17 knockdown inhibited the proliferation and induced the apoptosis of B-cell lymphoma cells, and the combination of SNHG17 and doxorubicin had a synergistic effect. SNHG17, miR-34a-5p, and ZESTE gene enhancer homolog 2 (EZH2) had common hypothetical binding sites, and the luciferase reporter assay verified that miR-34a-5p was the direct target of SNHG17, and EZH2 was the direct target of miR-34a-5p. The carcinogenic function of SNHG17 in the proliferation and apoptosis of DLBCL cells was partially reversed by a miR-34a-5p inhibitor. SNHG17 increases EZH2 levels by inhibiting miR-34a-5p. Our findings indicate SNHG17 as critical for promoting DLBCL progression by regulating

**Funding:** This study was supported by grants from the Natural Science Foundation of Guangxi (2018GXNSFBA281026, 2019GXNSFAA245086), the National Natural Science Foundation of China (82260042).the China Postdoctoral Science Foundation (2020M673555XB), and the "Future academic star" of Guangxi Medical University (WLXSZX22109). The funders had no role in study design, data collection and analysis, decision to publish, or preparation of the manuscript.

**Competing interests:** The authors have declared that no competing interests exist.

the EZH2 signaling pathway and sponging miR-34a-5p. These findings provide a new prognostic marker and therapeutic target for the prognosis and treatment of DLBCL.

## Introduction

Diffuse large B-cell lymphoma (DLBCL) is the most common type of non-Hodgkin's lymphoma, accounting for approximately 30–40% of non-Hodgkin's lymphoma cases. DLBCL is a highly invasive and heterogeneous malignant proliferative disease of the lymphatic system, comprising complex clinical manifestations, morphology, immune typing, and molecular genetic characteristics [1]. Recently, the incidence of DLBCL has gradually increased. For example, in China, the incidence of lymphoma is 2.9/100,000 [2]. Currently, chemotherapy drugs such as rituximab, cyclophosphamide, adriamycin, vincristine, and prednisone (R-CHOP) remain the cornerstone of DLBCL chemotherapy, but their curative effect is limited. Furthermore, the cure rate of adult DLBCL patients treated with chemotherapy alone is less than 40%, and the prognosis of older patients is poor. The main reason for this is tumor drug resistance [3]. Therefore, exploring the molecular mechanisms of DLBCL progression to develop more effective treatment strategies is very important.

Long non-coding RNA (lncRNAs) are RNA molecules with a length of more than 200 nucleotides. Although it lacks an open reading frame and does not have or has limited protein-coding ability [4, 5], many studies have shown that lncRNA can play a role in gene transcription and translation. The role affects the occurrence and development of tumors [6–8]. The malignant phenotype of tumors is closely related to various lncRNAs [9, 10]. lncRNAs can regulate the carcinogenic and tumor-suppressive pathways by competitively binding to miRNAs [11, 12] as competing endogenous RNAs (ceRNAs). For example, HNF1A-AS1 is induced by MYC and promotes glioma progression via miR-32-5p/SOX4 [13]. lncRNAARHGAP18 promotes tumor metastasis in hepatocellular carcinoma through miR-153-5p [14].

lncRNA small nucleolar RNA host genes (SNHGs), a subset of long RNA, are spliced from the exons of the primary RNA transcripts of SNHGs and transported to the cytoplasm. They are abnormally expressed in cancer and participate in cell proliferation, tumor progression, metastasis, and chemoresistance [15]. Various SNHGs have been reported to be involved in cancer progression [16–18]. For example, small nucleolar host gene 17 (SNHG17) can cause carcinogenesis in gastric cancer [19], colorectal cancer [20], non-small cell lung cancer [21], oral squamous cell carcinoma [22], and breast cancer [23]. However, the roles of SNHG17 in various tumors differ. In addition, the correlation between SNHG17 and DLBCL is unclear, and its potential mechanism has not been investigated. Therefore, understanding the expression pattern and regulatory mechanism of SNHG17 may help identify new prognostic markers and potential therapeutic targets for DLBCL.

MicroRNAs (miRNAs) are a small subclass of short non-coding RNA composed of approximately 22 nucleotides [24]. To date, miRNAs have been shown to regulate cellular activities associated with cancer development, including proliferation, apoptosis, and escape from anti-tumor immune responses [25, 26]. miRNAs target the 3'- untranslated region (3'-UTR) of mRNA through base pairing, blocking and degrading the translation of target genes and preventing the production of functional proteins. miRNAs are considered inhibitors of gene expression [27, 28]. For example, miR-21 inhibits breast tumorigenesis and angiogenesis by targeting the VEGF/VEGFR2/HIF-1 $\alpha$ axis [29]. microRNA-139 can inhibit the progression of liver cancer by negatively regulating KPNA2 [30]. miR-34a-5p has been investigated in ovarian cancer [31], cervical cancer [32], and head and neck squamous cell carcinoma [33] and is

considered a tumor suppressor gene that can inhibit cell proliferation. In this study, we aimed to verify the clinical significance of SNHG17 and explore its biological regulation in DLBCL cells. In particular, we used lncRNA–miRNA–mRNA interaction effects to determine potential functional mechanisms in DLBCL.

## Materials and methods

### Bioinformatics analysis

GSE10846 and GSE31312 datasets were downloaded from the Gene Expression Omnibus (GEO) database. The cases in both datasets are newly diagnosed as DLBCL, and their characteristics are shown in Table 1. Survival information was matched according to the datasets. A Cox regression model was constructed according to hazard ratio, $<1$ or $>1$, and $p<0.05$ as the screening conditions, low-risk and high-risk lncRNAs were shown. UCSC XENA (https:// xenabrowser.net/datapages/) database was processed in a unified way through the Toil process [34].The RNA-seq data in TPM (Translations per million reads) format were extracted from the diffuse large B-cell lymphoma patient tissues in The Cancer Genome Atlas and the corresponding normal tissues in GTEx. After log2 was transformed, the expression of SNHG17 in the samples was compared [35]. By reviewing studies and qPCR to verify the expression of SNHG17 in normal tissues (peripheral blood mononuclear cell; PBMC) and B-cell lymphoma cell lines, lncRNA SNHG17 was determined as the target lncRNA in this study.

### Construction of nomograph

A nomogram is a graphical display of clinical prediction models with intuitive advantages. The nomogram in this study was based on multivariate Cox regression analysis, which was scored according to the regression coefficient of each risk factor, and the scores of each risk factor were added together to obtain the total score to calculate the risk of outcome events. Data used to construct the nomogram were obtained from the GSE10846 dataset. In addition, SNHG17 mutations were screened to establish a prognostic model. The endpoint of the study was overall 3- and 5-year survival rates.

### Cell culture and transfection

PBMC and B lymphoma cell lines (Daudi, Ramos, DoHH2, Farage, and Raji) were purchased from Procell (Nanjing Cobioer Biosciences Co, LTD, China) and cultured in a 1640 Roswell

**Table 1. Clinical characteristics of DLBCL datasets(GSE10846 and GSE31312).**

|  | GSE10846 | | GSE31312 | |
|---|---|---|---|---|
| Age | ≤60 years | 188 | ≤60 years | 180 |
|  | >60 years | 226 | >60 years | 248 |
| Gender | male | 233 | male | 245 |
|  | female | 181 | female | 183 |
| Pathological Subtypes | GCB | 183 | GCB | 219 |
|  | non-GCB | 231 | non-GCB | 209 |
| Stage | I-II | 192 | I-II | 203 |
|  | III-IV | 222 | III-IV | 225 |
| ECOG | 0–1 | 296 | 0–1 | 350 |
|  | >1 | 118 | >1 | 78 |

Footnote: ECOG, Eastern Cooperative Oncology Group; GCB, Germinal center B-cell.

Park Memorial Institute (RPMI-1640) medium containing 10% fetal bovine serum. Human renal epithelial cells (239T) were purchased from the cell bank of the Chinese Academy of Sciences (Shanghai, China) and cultured in Dulbecco's modified Eagle medium (DMEM) containing 10% fetal bovine serum.All the above cells were cultured in DMEM/RPMI-1640 (Gibco, Waltham, MA, USA) supplemented with 10% fetal bovine serum (Gibco) and 100 U/mL penicillin and streptomycin (Gibco) at 37˚C in a 5% $CO_2$ incubator. The SNHG17 fragment was inserted into GV493 for knockdown (sh-SNHG17), and a blank GV493 vector (sh-NC) was used as a control. SNHG17 Short hairpin RNA (sh-RNA1#, sh-RNA2#, sh-RNA3#) and sh-NC were directly designed and synthesized by Jikai Gene Medical Technology (Shanghai, China). The miR-34a-5p simulant, negative control (NC) simulant, and miR-34a-5p inhibitor were obtained directly from GenePharma (China). Lipofectamine 3000 and Opti MEM (Invitrogen) were used to transfect cells.

## Real-time quantitative qPCR analysis

A TRIzol Kit (KeyGEN BioTECH, Nanjing, China) was used to lyse the tumor tissues and cells. Total RNA was extracted, and UV spectrophotometry (NanoDrop 2000; Thermo Fisher Scientific, Beijing, China) was used to determine the concentration and purity of total RNA. Total RNA was denatured at 65˚C for 15 min and cooled on ice. Each RNA sample was reverse-transcribed to cDNA at 37˚C for 15 min and then denatured at 85˚C for 5 s. Specific primers and ABI ViiA™7 PCR (Applied Biosystems; Thermo Fisher) were used for qPCR. The primer sequences used were as follows: SNHG17 forward, 5′-CGTGAATCTCTTGGTGGT GTTTGTG-3′; SNHG17 reverse, 5′-CTCTGGTGACGCTTCATGTGGTAG-3′. EZH2 forward, 5′-ATGATAAAGAAAGCCGCCCACCTC-3′; EZH2 reverse, 5′-TTCTGCTGTGCCCTTATC TGGAAAC-3′. GAPDH forward, 5′-CCCCGCTACTCC TCCTCCTAAG-3′; GAPDH reverse, 5′-TCCACGACCAGTTGT CCATTCC-3′. miR-34a-5p forward, 5′-AGCGCCTTGGCAGTG TCTTA-3′. RNU6B forward, 5′-CTCGCTTCGGCAGCACA-3′; RNU6B reverse, 5′-AAC GCTTVACGAATTTGCGT-3′. GAPDH was the internal reference control for SNHG17 and EZH2, and RNU6B was the internal reference control for miR-34a-5p. Real-time fluorescence quantitative PCR data were analyzed by the $2^{-\triangle\triangle Ct}$ method, and the $2^{-\triangle\triangle Ct}$ value represented multiple changes.

## Cell proliferation assay

DoHH2 cells ($1 \times 10^3$/mL) and Daudi cells ($1 \times 10^3$/mL) with logarithmic growth after transfection were resuspended and cultured in 96-well plates. B-cell lymphoma cells were treated with different doxorubicin (DOX) (Actavis Italy S.p.A) concentrations (0.03125, 0.0625, 0.125, 0.25, 0.5, and 1 μM) and cultured for 0, 24, 48, 72, and 96 h. Cell proliferation was measured using a Cell Counting Kit-8 (CCK-8) assay (Sangon Biotech, Shanghai) according to the manufacturer's instructions. Absorbance at 450 nm was measured using a microplate reader (Thermo Fisher).

## Cell apoptosis detection

DoHH2 cells ($2 \times 10^6$ /mL) and Daudi cells ($2 \times 10^6$ /mL) were cultured in a 6-well plate for 48 h and washed twice with PBS. Each sample and 195 μL staining buffer were mixed and stained with APC and 7-AAD labeled AnnexinV (BD Bioscience, San Jose, CA, USA) and incubated for 20 min. Cell apoptosis was detected using flow cytometry (Aceabio, USA).

After the experiment, nude mice were euthanized and tumors were completely removed. Fresh tumor tissue from each group was collected and ground at 4˚C to form a homogenate and filtered through a 300-mesh sieve; PBS was added to form a cell suspension for counting.

Cells ($1 \times 10^7$) were taken from each group and centrifuged at 377.325 g for 5 min. The supernatant was discarded and resuspended with PBS (1 mL). Then, the sample was centrifuged at 1500 rpm for 5 min before being washed twice in PBS. The supernatant was discarded, and the liquid was left at 50 μL. Upflow cytometry was used to detect the cell apoptosis rate (refer to the method for in vitro detection of cell apoptosis).

## Western blot analysis

The transfected DoHH2 and Daudi cells were lysed with RIPA buffer (Solarbio, China), and the total protein was quantified with a BCA protein content detection kit (Beyotime, Nanjing, China). Each sample (20 μg) was used for western blot analysis. Protein lysates were separated by 12% SDS sulfate-polyacrylamide gel electrophoresis (SDS-PAGE) and transferred to a 0.22 mm Immobilon-P membrane (Millipore, USA). The membrane was blocked for 1 h and incubated with a specific primary antibody overnight at 4˚C. The sample was washed three times with PBS, followed by incubation with a secondary antibody for 1 h at 25˚C, and the protein signal was detected using a chemiluminescence detection system (GE Healthcare, Chicago, IL, USA). The primary antibodies used in this study were as follows: anti-EZH2 (1:2000, ab283270, Abcam), anti-PARP (1:1500, ab229756, Abcam), and GAPDH antibody (1:10000, Cell Signaling Technology, USA), followed by incubation with peroxidase-labeled goat anti-mouse IgG (Cell Signaling Technology) at 1:5000 dilution for 1 h at room temperature. GAPDH was used as the internal control.

## Xenotransplantation tumor formation

The animal experiments were conducted in accordance with the guidelines for animal care and use of Guangxi Medical University and were approved by the animal experiment ethics committee of the animal center of Guangxi Medical University (SCXKGUI2020-0003). DoHH2 cells ($1 \times 10^7$/mL) transfected with sh-NC and sh-SNHG17 were subcutaneously injected into the backs of BALB/c male nude mice for approximately four weeks (n = 4). Tumor volume was recorded every four days after implantation. After 28 days, the tumor weight was measured.

## Gene set enrichment analysis (GSEA)

GSEA 4.0.3 software and Java 8 operating environment were downloaded from the GSEA website (https://www.gsea-msigdb.org/gsea/downloads.jsp). These software packages performed GEO enrichment analysis of clinical RNA sequencing data (GSE10846) and SNHG17 knockdown cell model RNA sequencing data. Two groups of data were prepared for input into the model: ① Clinical data were prepared according to the median of SNHG17 expression in the GEO database and were divided into a low SNHG17 expression gene matrix and a high SNHG17 expression gene matrix. ② Cellar data involved taking the scrambled RNA sequencing data of DoHH2 and Daudi cells as the input data of the control group and the RNA sequencing data of KD of DoHH2 and Daudi cells as the input data of the knockdown group. The above two groups of data were imported into the GSEA software in turn, and then c2 All V7.5 Symbols Gml [calibrated] was run and GSEA was performed, with each analysis repeated 1000 times according to the default weighted enrichment statistical method, which included four key statistics: normalized enrichment score (NES), enrichment score, normalized P-value, and false discovery rate. The data were filtered using the following conditions: the absolute value of NES was greater than or equal to 1.7, and the normalized P-value was less than or equal to 0.0001. The intersection of the two groups of enriched signal pathways was taken, the

Wayne diagram was created according to the intersection, and the common signal pathways of clinical and cell lines are shown in a bar graph.

### Dual-luciferase reporter assay

To verify the interaction between SNHG17 and miR-34a-5p, miR-34a-5p, and EZH2, a reporter vector pGL3 containing wild-type and mutant binding sites of miR-34a-5p was constructed (SNHG17WT and SNHG17MUT, EZH2WT, and EZH2MUT, respectively). miR-34a-5p simulant or NC simulant and wild-type/mutant pGL3 plasmids were co-transfected into 239T cells using Lipofectamine 3000 (Invitrogen). Luciferase activity was detected using a double-luciferase reporter gene detection system (Promega).

### Statistical analysis

All experiments were repeated three times, and bioinformatics and experimental data were analyzed using the R program (version 3.41). Univariate and multivariate Cox regression analyses were used to analyze the effects of lncRNA expression on patient survival. ROC curves were plotted using the pROC software package. The comparison between groups was evaluated using Student's t-test (between two groups) and one-way ANOVA (between three or more groups). The data are expressed as mean ± standard deviation (SD), and $P < 0.05$ was considered to indicate significance.

## Results

### Screening the lncRNAs related to the prognosis of DLBCL patients in the GEO database

We performed Cox regression analysis on the annotated lncRNAs in GSE10846 and GSE31312 datasets; $P < 0.05$ was the screening condition. In the hazard ratio < 1 group, there were 61 lncRNAs in GSE10846 and 220 lncRNAs in GSE31312. We took the intersection of the two datasets and selected 16 lncRNAs with a good prognosis and low risk in patients with DLBCL (Fig 1A and 1C). In the hazard ratio> 1 group, there were 178 lncRNAs in GSE10846 and 55 lncRNAs in GSE31312. We considered the intersection of the two datasets and selected 13 lncRNAs with poor prognosis and high risk in patients with DLBCL (Fig 1B and 1D).

### Prognosis analysis of SNHG17

In the dataset GSE10846, we further analyzed the association between SNHG17 and clinical parameters and survival. The Kaplan–Meier survival curve analysis was performed according to the SNHG17 expression order. SNHG17 had low expression at 0–25% and high expression at 75%–100% (Fig 2A). To check the robustness of the prognostic model, we used the same risk score formula obtained from the training cohort and calculated the risk scores of all patients in the test cohort, which were divided into high- and low-risk groups using the same threshold as the training cohort. We performed the same survival analysis as that for the training cohort. Consistent with the results of the training cohort, the OS of high-risk patients was shorter than that of low-risk patients in the test cohort (Fig 2B, log-rank test, $P < 0.0001$). Time-dependent ROC analysis showed that the AUC of SNHG17 was 0.741 (95% CI: 0.669–0.814) and 0.725 (95% CI: 0.637–0.812) at 3 and 5 years, respectively (Fig S1 in S1 Data). The classification based on the risk score of the validation cohort also showed similar results (Fig 2C, log-rank test, $P = 0.0018$; AUC of 3 years: 0.654, 95% CI: 0.530–0.779; AUC of 5 years: 0.653, 95% CI: 0.508–0.798; Fig S2 in S1 Data). Based on the construction of the nomogram of the Cox regression model, we fitted the complete set of datasets with Cox regression of single

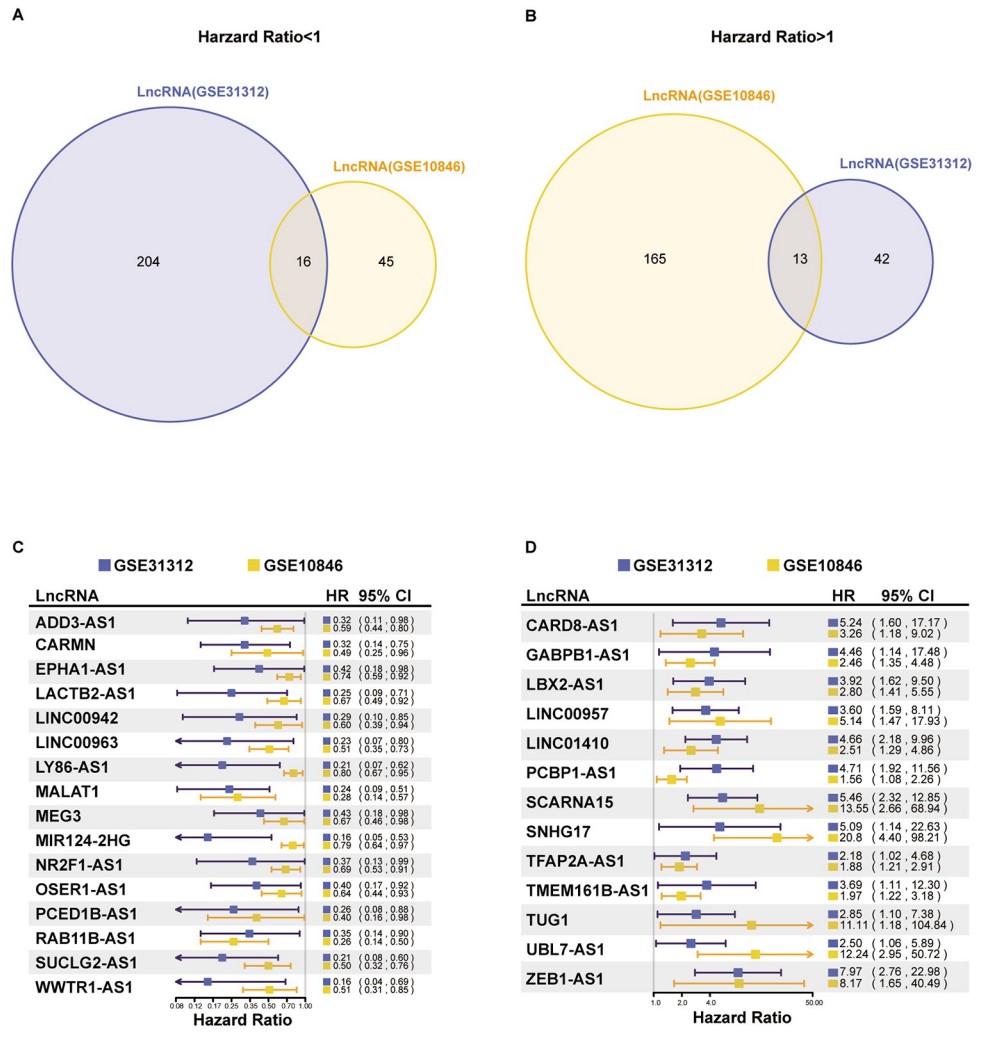

**Fig 1. Screening the lncRNAs related to DLBCL prognosis and survival.** (A, B) Venn diagram of GSE10846 and GSE31312 datasets. (C) Forest diagram of DLBCL patients with good prognosis and low-risk lncRNAs. (D) Forest diagram of DLBCL patients with poor prognosis and high-risk lncRNAs.

and multiple factors. In univariate and multivariate survival analyses, the trend was consistent with the previous validation and test sets (Fig 2D and 2E). We constructed a nomogram to provide a quantitative method for clinicians to predict the possibility of individual survival, in which lncRNAs were integrated with clinicopathological independent survival risk factors (including age, tumor stage, LDH level, and ECOG score) (Fig 2F). The 3- and 5-year deviation calibration lines in the calibration map were very close to the ideal curve (45˚ line), indicating that the consistency between prediction and observation was very good (Fig S3 in S1 Data).

## SNHG17 is highly expressed in DLBCL patients' tumor cells and in B-lymphoma cell lines

The UCSC XENA database was used to compare the expression of SNHG17 between normal control tissues and tumor samples after the RNA-seq data were log2 transformed. The results showed that the expression of SNHG17 in diffuse large B-cell lymphoma patients' tissues was higher than that in normal control tissues (P < 0.001) (Fig 2G).To verify the expression of

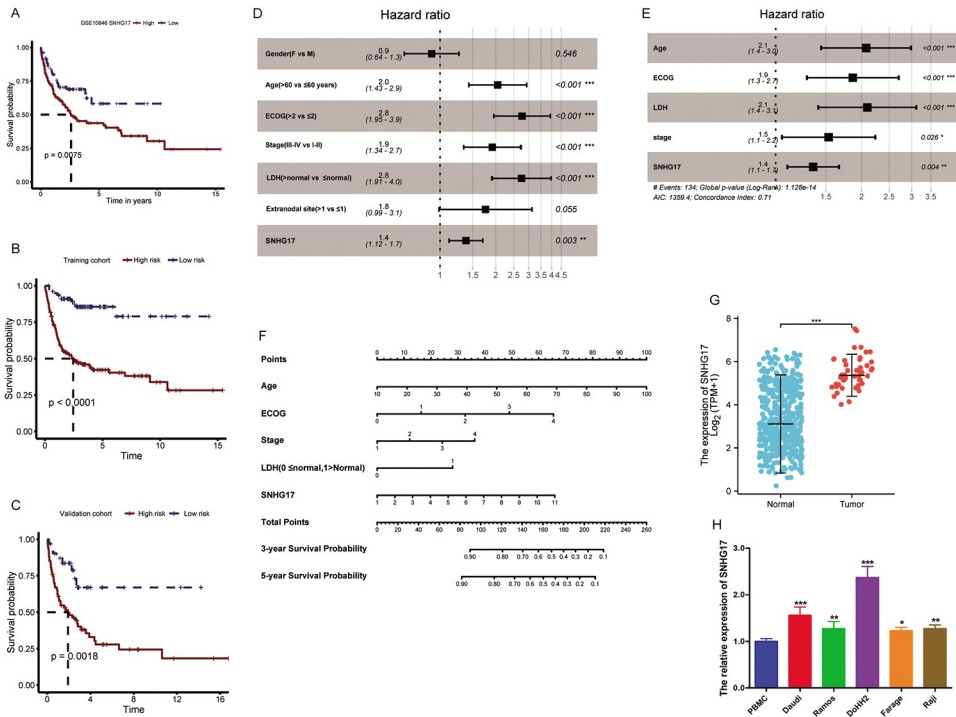

**Fig 2. SNHG17 is highly expressed in DLBCL patients' tumor cells and in B-lymphoma cell lines.** (A) Survival analyses of GSE10846 (n = 420) in DLBCL patients with high and low expression of SNHG17.(B, C) Kaplan–Meier survival analysis. (D, E) Uniform and multivariate expression analyses (all bars correspond to the 95% confidence intervals). (F) Nomogram was used to predict the OS of patients. (G) Expression of SNHG17 in DLBCL patients' tumor and normal tissues.The sequencing data for normal cells and diffuse large B-cell lymphoma were obtained from UCSC XENA, which we have described in the methods section. (H) qPCR shows the expression of SNHG17 in lymphoma cell lines (Daudi, Ramos, DoHH2, Farage, and Raji) and peripheral blood mononuclear cells (PBMC). *P < 0.05, **P < 0.01, ***P < 0.001.

SNHG17 in lymphocytes, qPCR was used to detect the expression of SNHG17 in PBMC and various lymphoma cell lines. qPCR results showed that the expression of SNHG17 in lymphoma cell lines (Daudi, Ramos, DoHH2, Farage, and Raji) was higher than that in PBMCs, with the highest expression in DoHH2 cells (Fig 2H). Then, we constructed DoHH2 and Daudi cells with SNHG17 knocked down by stably transfecting sh-SNHG17 in DoHH2 and Daudi cells. qPCR results showed that sh-SNHG17#2 knockdown effect in both cells was lower than that of sh-SNHG17#1/3 Figs S4 and S5 in S1 Data. Therefore, we selected sh-SNHG17#2, with the most significant downregulation for subsequent analysis. The sh-SNHG17#2 used in subsequent experiments is called sh-SNHG17. The results showed that SNHG17 was upregulated in DLBCL patient tissues and cell lines and that higher SNHG17 levels were positively correlated with a lower survival rate.

## SNHG17 knockdown inhibits proliferation and promotes apoptosis of DLBCL cells

We further explored the biological functions of SNHG17 in DLBCL. When sh-SNHG17 was transfected into DoHH2 and Daudi cells, we found that the expression of SNHG17 was significantly decreased. The CCK-8 assay showed that the knockdown of SNHG17 inhibited the proliferation of DoHH2 and Daudi cells, and there was a synergistic effect with DOX(0.125 μM) (Fig 3AI and 3AII). In addition, flow cytometric analysis showed that: sh-SNHG17 significantly

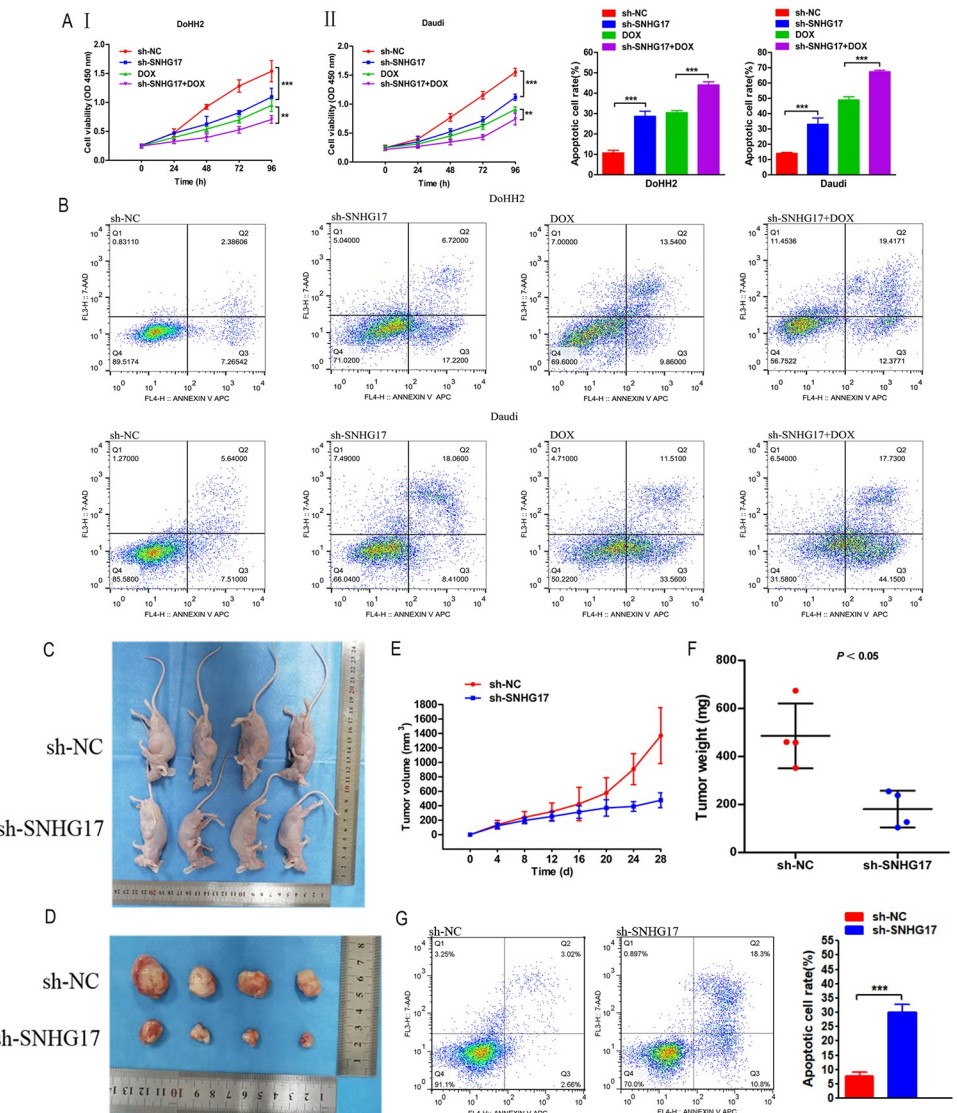

**Fig 3. SNHG17 can promote the growth of DLBCL cells in vitro and in vivo.** (AI-lI) CCK-8 experiment confirmed the proliferation ability of DoHH2 and Daudi cells, and the concentration of DOX was 0.125 μM. (B) Flow cytometry was used to analyze the apoptosis of DoHH2 and Daudi cells. (C-F) DoHH2 cells transfected with sh-NC or sh-SNHG17 were inoculated into the back of nude mice. The tumor planting diagram (C), picture (D), growth curve (E), and weight (F) are shown. (G) Flow cytometry was used to analyze the apoptosis of transplanted tumor cells. The data are expressed as mean ± SEM* $P < 0.05$, **$P < 0.01$, ***$P < 0.001$. n = 4 mice per group.

increased the apoptosis rate of DoHH2 and Daudi cells (Fig 3B).Sh-SNHG17 increased the proteolytic cleavage of 116 kDa PARP to its 89 kDa fragment, which is a hallmark of apoptosis in DoHH2 and Daudi cells.Sh-SNHG17 appears to enhance doxorubicin sensitivity (Fig S6 in S1 Data). In conclusion,sh-SNHG17 inhibited proliferation and promoted apoptosis of DLBCL cells.

## SNHG17 knockdown inhibits tumor growth in vivo

We also investigated the role of SNHG17 in DLBCL progression in vivo. First, we stably transfected DoHH2 cells with sh-SNHG17 or sh-NC. The sh-NC group cells ($1 \times 10^7$/mL) and sh-

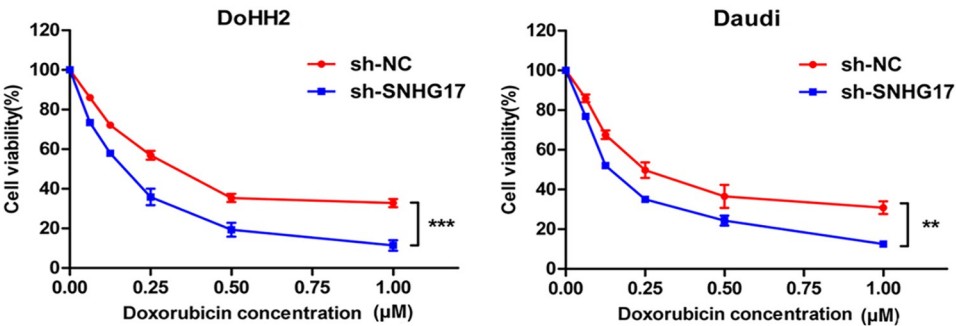

**Fig 4. SNHG17 knockdown enhanced the sensitivity of DLBCL cells to doxorubicin.** Knockdown of SNHG17 enhanced the sensitivity of lymphoma cells to doxorubicin. *P < 0.05, **P < 0.01, ***P < 0.001.

SNHG17 group cells ($1 \times 10^7$/mL) were inoculated subcutaneously into BALB/c nude mice. During the 28-day modeling process, sh-SNHG17 mice showed smaller DLBCL tumors (Fig 3C and 3D), and the sh-SNHG17 group also had smaller average volumes and lighter tumor weights (Fig 3E and 3F).Through flow cytometric analysis, we examined the apoptosis of transplanted tumor cells (Fig 3G).In conclusion, these results suggested that SNHG17 plays a carcinogenic role in DLBCL.

## SNHG17 knockdown can enhance the sensitivity of DLBCL cells to DOX

To investigate the relationship between SNHG17 and the chemosensitivity of lymphoma cells, we determined the sensitivity of cells to DOX using the CCK-8 method. Our results showed that SNHG17 knockdown significantly reduced the semi-inhibitory DOX concentration in DoHH2 and Daudi cells (Fig 4).

## GSEA of SNHG17 in B-cell lymphoma

GSEA was used to analyze the signaling pathways involved in the enrichment of low SNHG17 expression in clinical samples and SNHG17 knockdown in cell line samples. The results showed that SNHG17 participates in many key pathways in the occurrence and development of tumors. A total of 507 signal pathways were enriched in the clinical sample data, and 911 signal pathways were enriched in the cell sample data, of which 21 signal pathways were jointly involved in the above clinical samples and cell line samples (Fig 5A). The NES values of the same signal path of the two samples were added, sorted according to the absolute value of the NES, and displayed with a bar graph. The top three signal paths include EZH2_ Targets, MYB_ Targets, and Natural_ Killer_ Differentiation (Fig 5B).

## Searching for bridge biomolecules between SNHG17 and EZH2

Using Diana tools (https://diana.e-ce.uth.gr/lncbasev3/interactions), 21 putative binding sites of miRNA and SNHG17 were predicted, such as miR-101-3p, miR-34a-5p, and miR-17-5p. Using Starbase v2.0 (https://starbase.sysu.edu.cn/starbase2/browseIntersectTargetSite.php), five putative binding sites of miRNA and EZH2 were predicted, such as miR-34a-5p and miR-101-3p. Considering the intersection of the miRNAs predicted by the above two databases, it can be concluded that the two miRNAs may bind to SNHG17 and EZH2 simultaneously, as shown in the Venn diagram (Fig 5C). Furthermore, qPCR showed that the expression of miR-34a-5p and miR-101-3p was significantly downregulated in DLBCL cells compared with that

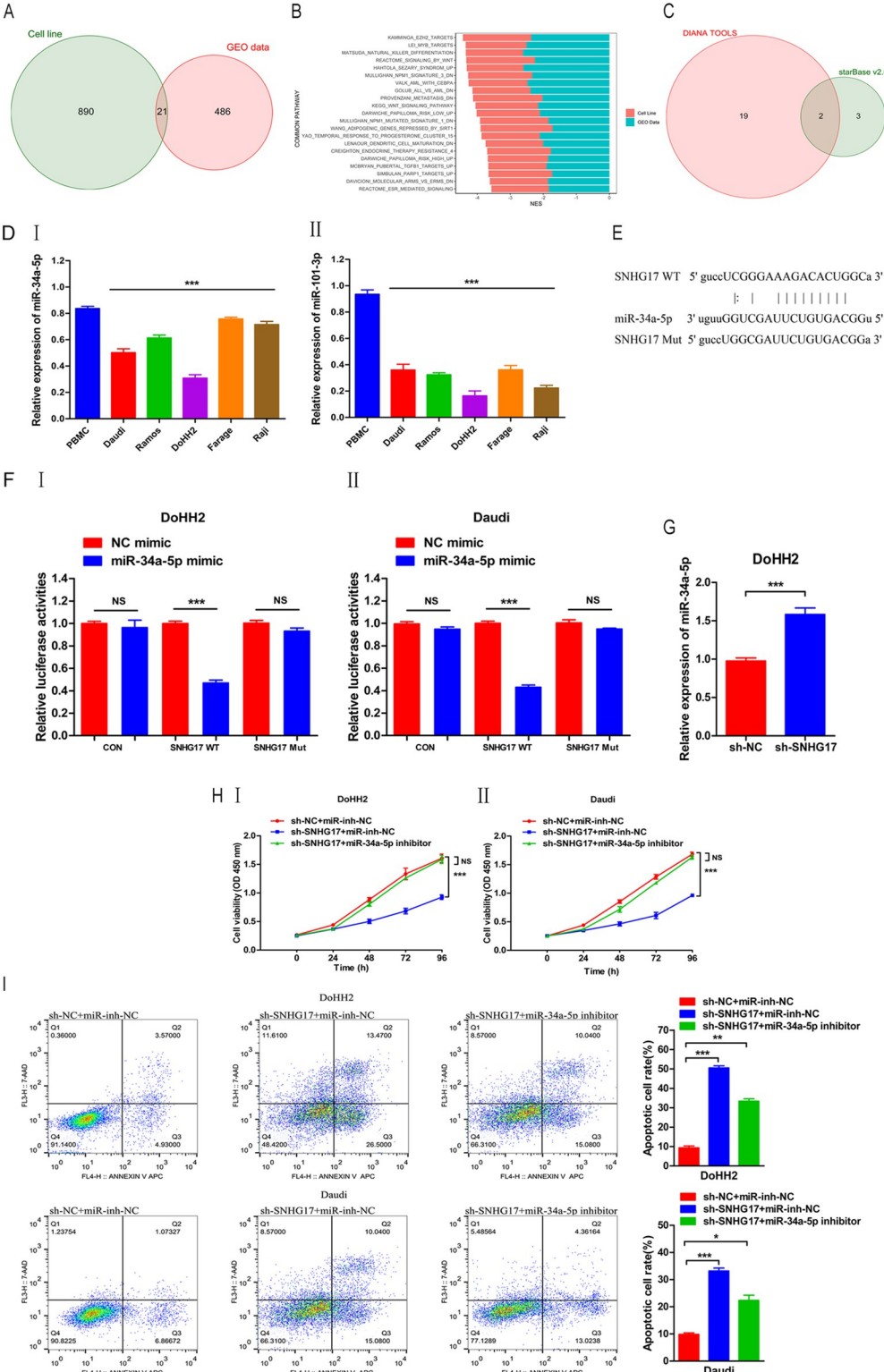

**Fig 5. miR-34a-5p is the direct target of SNHG17; miR-34a-5p inhibitor reverses the inhibitory effect of SNHG17 knockdown on lymphoma cells.** (A) Signal pathways enriched from clinical and cell line lymphoma samples were taken at the Venn diagram intersection. (B) Bar graph of common signal pathways in clinical and cell line samples. (C) Venn diagram of miRNA binding to SNHG17 and EZH2 at the same time. (DI-II) Expression of miR-34a-5p and miR-101-3p in lymphoma. (E) Binding sites of miR-34a-5p and SNHG17 were predicted by bioinformatics analysis. (FI-II)

Double luciferase reporter was used to confirm the binding. (G) Expression of miR-34a-5p decreased in lymphoma cells and was negatively correlated with SNHG17. (HⅠ-ⅡⅠ) miR-34a-5p inhibitor reversed the effect of SNHG17 knockdown on cell proliferation. (I) miR-34a-5p inhibitor reversed the effect of SNHG17 knockdown on cell apoptosis. *P < 0.05, **P < 0.01, ***P < 0.001.

in PBMC (Fig 5DⅠ and 5DⅡ). Since the expression of miR-101-3p was too low in lymphoma cells, we verified the functional expression of miR-34a-5p in DLBCL cells.

### miR-34a-5p is the direct target of SNHG17; miR-34a-5p inhibitor reversed the inhibitory effect of SNHG17 knockdown on lymphoma cells

According to miRDB (http://mirdb.org/) and the MiRbase database (https://mirbase.org/index.shtml), it is predicted that SNHG17 is a regulator of miR-34a-5p. We found that SNHG17 and miR-34a-5p shared a common hypothetical binding site (Fig 5E). The double-luciferase reporter assay confirmed this prediction. Transfection of miR-34a-5p decreased the luciferase activity of SNHG17-WT cells (P < 0.01), but no significant effect was observed in SNHG17-MUT cells (Fig 5FⅠ and 5Ⅱ).qPCR results showed that compared with PBMC, the expression of miR-34a-5p in DLBCL cells was significantly downregulated (Fig 5D). In addition, SNHG17 knockdown increased the expression of miR-34a-5p in DoHH2 cells (Fig 5G). This result indicated that miR-34a-5p is negatively regulated by SNHG17 in this cell line.

### miR-34a-5p inhibitor can reverse the effects of SNHG17 knockdown on lymphoma cell proliferation and apoptosis

We investigated whether SNHG17 regulates DLBCL progression by inhibiting miR-34a-5p. CCK-8 results showed that downregulating SNHG17 in DoHH2 and Daudi cells could significantly inhibit cell proliferation, while miR-34a-5p inhibitor could partially rescue cell proliferation (Fig 5HⅠ and 5HⅡ). We also investigated the role of SNHG17/miR-34a-5p in apoptosis. Compared with the control group, DoHH2 and Daudi cells transfected with sh-SNHG17 showed a higher apoptosis rate, while the miR-34a-5p inhibitor significantly reversed the DoHH2 and Daudi cell apoptosis caused by SNHG17 knockdown (Fig 5I). Therefore, SNHG17 may regulate the proliferation and apoptosis of DLBCL cells by negatively regulating miR-34a-5p.

### EZH2 is highly expressed in lymphoma cells and is negatively regulated by SNHG17

EZH2 and miR-34a-5p shared a common putative binding site (Fig 6A). The double luciferase reporter assay confirmed this prediction. Transfection of miR-34a-5p decreased the luciferase activity of EZH2-WT cells (P < 0.01), but no significant effect was observed in EZH2-MUT cells (Fig 6BⅠ and 6BⅡ). In addition, overexpression of miR-34a-5p decreased EZH2 expression, whereas EZH2 was highly expressed in lymphoma cells (Fig 6CⅠ and 6CⅡs). These results indicated that miR-34a-5p negatively regulates EZH2. To understand whether miR-34a-5p affects the EZH2 pathway, we detected EZH2 protein expression after miR-34a-5p overexpression in lymphoma cells by western blotting (Fig 6DⅠ and 6DⅡ). The miR-34a-5p inhibitor significantly reversed the inhibitory action of EZH2 in DoHH2 and Daudi cells caused by SNHG17 knockdown (Fig 6E and 6F).

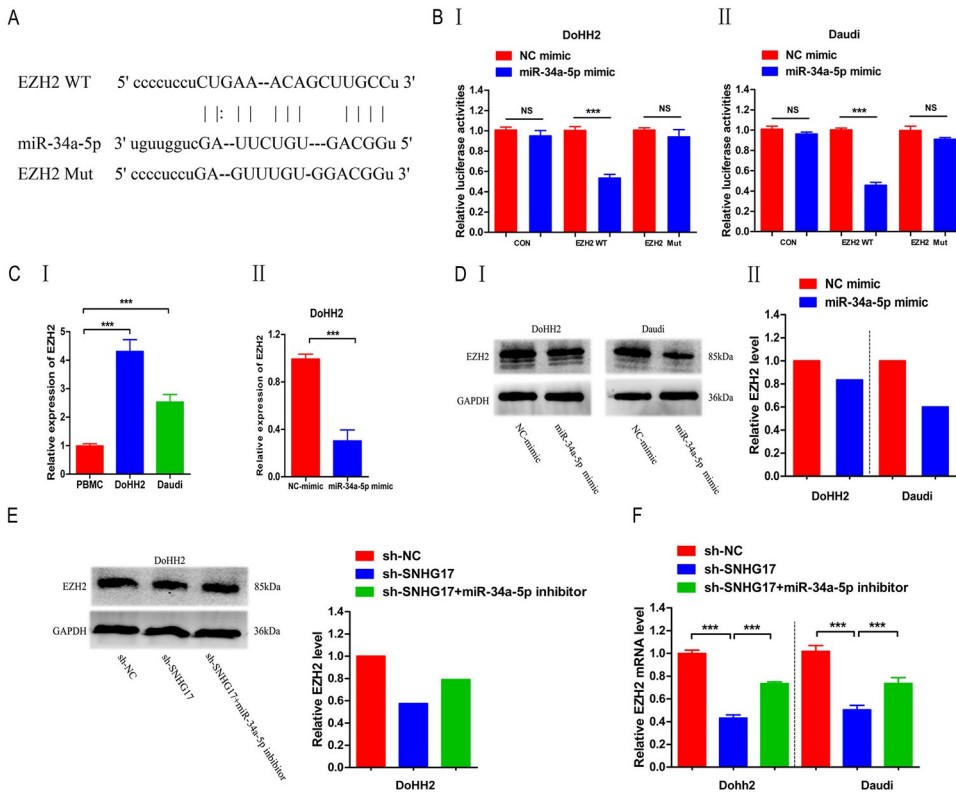

**Fig 6. miR-34a-5p affects the EZH2 protein pathway.** (A) Binding sites of EZH2 and miR-34a-5p were predicted through bioinformatics analysis. (BI-II) Double luciferase reporter experiment confirmed this prediction. (CI-II) Expression of EZH2 was increased in lymphoma cells and negatively correlated with miR-34a-5p. (DI-II) Overexpression of miR-34a-5p resulted in significant inhibition of the EZH2 signaling pathway. (E, F) miR-34a-5p inhibitor significantly reversed the EZH2 protein inhibition caused by SNHG17 knockdown. *P < 0.05, **P < 0.01, ***P < 0.001.

## Discussion

DLBCL is a group of B-cell lymphomas with strong heterogeneity in molecular pathology and cytogenetics. Pathologically, it is characterized by a diffuse distribution of malignant large B lymphocytes, and clinically, it shows an aggressive course. Recently, radiotherapy, chemotherapy, and immune checkpoint inhibitors have been used in clinical treatment; however, the prognosis of lymphoma remains poor. Although some mechanisms have been reported, the molecular characteristics of the pathogenesis of most lymphomas remain unclear. Many studies have shown that lncRNAs play important roles in the pathogenesis and development of many cancers [36]. Therefore, it is important to elucidate the molecular mechanisms of prognosis-related lncRNAs and explore potential therapeutic targets.

In this study, through high-throughput data and bioinformatics analysis, we found that SNHG17 was highly expressed in lymphoma tissues and cells and negatively correlated with the survival of patients with lymphoma. Loss of function analysis showed that SNHG17 knockdown could inhibit the proliferation of lymphoma cells (DoHH2 and Daudi) and synergistically affect DOX by inhibiting cell proliferation. SNHG17 knockdown also promotes cell apoptosis. In addition, we demonstrated for the first time that SNHG17 knockdown could enhance the sensitivity of lymphoma cells to DOX. However, as an oncogene involved in tumor progression, the mechanism by which SNHG17 is involved in lymphoma progression

remains unclear. According to previous studies, lncRNAs can interfere with miRNA–mRNA interactions by adsorbing miRNAs. miRNAs participate in various biological processes by cleaving target gene mRNA or inhibiting sequence-specific translation, including cell growth, differentiation, development, and apoptosis [37, 38]. For example, lncRNA SNHG17 promotes the progression of pancreatic cancer through cross-dialogue with miR-942 [39], and SNHG17 promotes the progression of cervical cancer by targeting microRNA-375-3p [40]. Studies have shown that tumor-related fibroblasts promote the proliferation and metastasis of oral cancer cells through exosome-mediated paracrine miR-34a-5p [41]. LncRNA 1700020I14Rik reduces cell proliferation and fibrosis via the miR-34a-5p/Sirt1/HIF-1α signaling pathway in diabetic nephropathy [42]. Our results showed that miR-34a-5p has a strong affinity for SNHG17. Overexpression of miR-34a-5p inhibited proliferation and promoted apoptosis of DLBCL cells, while SNHG17 knockdown and co-expression of miR-34a-5p attenuated this effect, indicating that SNHG17 could affect the proliferation of DLBCL cells by adsorbing miR-34a-5p. Bioinformatics and luciferase reporter gene analyses showed that SNHG17 played a carcinogenic role in DLBCL as a miRNA sponge, and there was an interaction between SNHG17 and miR-34a-5p. These results strongly suggested that miR-34a-5p is a direct target of SNHG17. However, the molecular mechanism by which miR-34a-5p inhibits tumor growth remains unclear.

In this study, we predicted the putative binding sites of miR-34a-5p and EZH2 using bioinformatics analysis, and this hypothesis was verified through luciferase reporter gene analysis. Early studies have found that EZH2 mutations are associated with Bcl-2 rearrangement in follicular lymphoma. Béguelin et al. investigated the relationship between DLBCL and Bcl-2 and found that mutant EZH2 had a stronger inhibitory effect on differentiating B-cell lymphoma cells. If it interacts with Bcl-2, it promotes the malignant transformation of germinal heart B cells. Combined with a Bcl-2 inhibitor, it can enhance the effect of EZH2 inhibitors [43, 44]. EZH2 is a catalytic subunit of the PRC2 protein complex that mediates the transcriptional silencing of target genes through 2/3 subunit methylation of H3K27 (H3K27me2/3) [45, 46]. Many studies have shown that EZH2 can activate the transcription of some lncRNAs in tumor cells [47] and plays a key role in the occurrence, progression, proliferation, and apoptosis of tumors [48]. Relevant studies have found that EZH2 is significantly overexpressed in lymphoid malignancies [49], gastric cancer [50], and uveal melanoma [51]. These results suggest that EZH2 mediates the regulation of SNHG17 during the immune escape and progression of DLBCL cells. However, the interaction between EZH2 and miR-34a-5p in malignant tumors has not yet been reported. Our study confirmed for the first time that EZH2 was upregulated in lymphoma cells and negatively correlated with miR-34a-5p and proposed that the upstream target gene SNHG17 regulates the downstream EZH2 signaling pathway by adsorbing miR-34a-5p, which is probably the reason for the resistance of lymphoma cells to DOX. SNHG17 knockdown inhibited EZH2 expression, while miR-34a-5p overexpression weakened this trend, suggesting that the SNHG17/miR-34a-5p axis may affect the biological function of lymphoid cancer cells and tumor development by regulating the activity of the EZH2 signaling pathway. Therefore, SNHG17 regulates the proliferation, apoptosis, and drug sensitivity of lymphoma cells by interacting with miR-34a-5p/EZH2.

In conclusion, our study found that SNHG17, a ceRNA, upregulated EZH2 expression and regulated the proliferation and apoptosis of DLBCL cells by adsorbing miR-34a-5p. Notably, these findings may provide a theoretical basis for SNHG17 as a new therapeutic target for the treatment of DLBCL. However, the main limitations of the current study are the lack of in vivo data and clinical evidence, which require further research.

## Supporting information

**S1 Data. SNHG17 is an lncRNA highly expressed in DLBCL.** (S1, S2) Time-dependent ROC curve analysis of Cox regression model in the test cohort. (S3) Nomograph calibration chart of 3- and 5-year OS probability.(S4) The knockdown efficiency of SNHG17 in DoHH2 cells was detected by qPCR. (S5) The knockdown efficiency of SNHG17 in Daudi cells was detected by qPCR.(S6) Apoptosis-associated protein PARP was examined using western blotting.
*P < 0.05, **P < 0.01, ***P < 0.001.
(JPG)

**S2 Data.**
(XLS)

**S1 Raw images.**
(PDF)

## Acknowledgments

We are grateful to the donors for their contributions to this study.

## Author Contributions

**Conceptualization:** Shengjuan Lu, Chao Rong, Sha He, Chengcheng Liao.

**Data curation:** Shengjuan Lu, Guojun Mo.

**Formal analysis:** Shengjuan Lu, Yuanhong Li, Chengcheng Liao.

**Funding acquisition:** Xiaoxun Xie, Chengcheng Liao.

**Investigation:** Xiaoxun Xie, Chengcheng Liao.

**Project administration:** Jie Sun, Dani Zhong.

**Resources:** Danqing Lei, Xiaohong Tan, Chengcheng Liao.

**Software:** Guojun Mo, Guodi Ou.

**Supervision:** Qing Ke, Xiaoxun Xie, Chengcheng Liao.

**Validation:** Hong Cen.

**Visualization:** Danqing Lei.

**Writing – original draft:** Shengjuan Lu, Lin Zeng, Guojun Mo.

**Writing – review & editing:** Shengjuan Lu, Guojun Mo, Hailian Wu, Qingmei Zhang.

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
