## [Decision Letter · Decision Letter 0]

21 Aug 2023

PONE-D-23-18237Long non-coding RNA SNHG17 may function as a Competitive Endogenous RNA in diffuse large B-cell lymphoma progression by sponging miR-34a-5pPLOS ONE

Dear Dr. Liao,

Thank you for submitting your manuscript to PLOS ONE. After careful consideration, we feel that it has merit but does not fully meet PLOS ONE’s publication criteria as it currently stands. Therefore, we invite you to submit a revised version of the manuscript that addresses the points raised during the review process.

We look forward to receiving your revised manuscript.

Kind regards,

Lucia Magnelli

Academic Editor

PLOS ONE

Reviewers' comments:

Reviewer's Responses to Questions

**Comments to the Author**

1. Is the manuscript technically sound, and do the data support the conclusions?

Reviewer #1: Yes

Reviewer #2: Yes

2. Has the statistical analysis been performed appropriately and rigorously? 

Reviewer #1: Yes

Reviewer #2: Yes

3. Have the authors made all data underlying the findings in their manuscript fully available?

Reviewer #1: Yes

Reviewer #2: Yes

4. Is the manuscript presented in an intelligible fashion and written in standard English?

Reviewer #1: Yes

Reviewer #2: No

5. Review Comments to the Author

Reviewer #1: The authors Lu et al., have determined the role of LncRNA SNHG17 in the progression of diffuse large B-cell lymphoma (DLBCL). They have shown that SNHG17 is a ceRNA and acts as a sponge for miR-34a-5p and regulates the EZH2 signaling pathway, and have identified SNHG17 as a potential therapeutic target for DLBCL.

Minor Comments:

The authors should proof read for grammatical and typographical errors.

Line 45, Page 9: miR-34a-5p is typed twice. Please rephrase to convey the interaction between SNHG17 and miR-34a-5p; and miR-34a-5p and EZH2.

Reviewer #2: This article is interesting and deserves to be published but needs some revisions. In general, English corrections must be made in the drafting of the English form and the organization of the text must be improved, making it more fluid and avoiding repetitions. Legends to figures needs more care. In the introduction section the aim of the paper must be better enucleated. More specific points are in the attached file.

6. PLOS authors have the option to publish the peer review history of their article (what does this mean?). If published, this will include your full peer review and any attached files.

Reviewer #1: No

Reviewer #2: **Yes: **LUCIA MAGNELLI

---

## [Author Response · Author response to Decision Letter 0]

13 Oct 2023

99 GSE10846 and GSE31312 datasets were downloaded from the Gene Expression Omnibus (GEO)

100 database. describe at least the main features of the chosen datasets: i.e. disease, treatment, number of subjects

Response：Firstly, we are so sorry for not mentioning this part. We have added the main characteristic of the GSE10846 and GSE31312 datasets in the methodology section of this paper, including Pathological Subtype, ECOG, Stage, Gender, and Age(they are showed in Table 1). 

123 serum. The cells were cultured in DMEM/ RPMI-1640 (Gibco, Waltham, MA, USA) supplemented

124 with 10% fetal bovine serum (Gibco) and 100 U/mL penicillin and streptomycin (Gibco) at 37 °C in

125 a 5% CO2 incubator. Which kind of cells?

Response：We are very sorry that our wording was not rigorous enough. We thank you for helping us identify mistake, and we have made correction according to the Reviewer ' s comments . These cells include PBMC cells, B lymphocyte cell lines (Daudi, Ramos, DoHH2, Farage, and Raji), and 239T cells mentioned above.

168 detected using a chemiluminescence detection system (GE Healthcare, Chicago, IL, USA). The

169 primary antibodies used in this study were as follows: anti-EZH2 (1:2000,Abcam Technology Inc.),

170 anti-PARPparp (1:1500,Abcam Technology Inc.), and anti-caspase-3 (1:1500,Abcam Technology Inc.) and Specify the name of ab clones.

Response： Thank you for the your suggestion. We have added the names of ab clones of anti EZH2, anti PARP in the method section of our paper.

253 3.3 SNHG17 is highly expressed in DLBCL patients’ tumor cells and in B-lymphoma cell lines (or DLBCL cell lines, as you call them in the latest part of the paper)

Response： Thank you for helping us to correct. We have made corrections based on the your comments.

SNHG17 is highly expressed in DLBCL patients’ tissues and cell lines

541 Figure 2. SNHG17 is highly expressed in DLBCL patients’ tumor cells and in B-lymphoma cell lines. (A) Survival

Response： We have made the corrections based on the reviewer's comments.

542 analyses of GSE10846 (n=70) DLBCL patients with high and low expression of SNHG17 in our cohort.

Response： Thank you for pointing out and making corrections. We have fixed them.

545 patients with OS. (G) Expression of SHG17 in DLBCL patients’ tumor and normal tissues, tumor: diffuse

546 large B-cell lymphoma. Explain better they are from CGA

Response：Thanks. As Reviewer suggested that the content of our statement is not complete enough. We have revised the legend. Also, we have described this in detail in the methods section.

261 constructed DoHH2 and Daudi cells with SNHG17 knocked down by stably transforming transfecting sh-

-Fig 2 from I to J and its description are neither pertinent to point 3.3., nor relevant at all. At most you can enter them in supplementary data. The denomination sh-SNHG17 can be introduced elsewhere, such as in the next paragraph or in “cell culture and transfection”

Response：We are so sorry for our incorrect writing and inappropriate data formatting. Thank you for pointing out and making revisions for us. We have revised the data layout based on the reviewer's suggestions, deleted the image from Fig 2 from I to J, and included them in the supplementary data.

274 of DoHH2 and Daudi cells (Fig. 3B). Western blotting showed SNHG17 knockdown increased parp

275 protein levels in DoHH2 and Daudi cells (Fig. 3C). In conclusion, SNHG17 knockdown inhibited

276 proliferation and promoted apoptosis of DLBCL cells.

Response：Thank you for pointing out this problem. We have replaced SNHG17 knockdown with sh-SNHG17.

283 Through qPCR and western blot detection, the mRNA and protein levels of caspase-3 decreased and

284 parp mRNA increased in SNHG17-knockdown tumor tissues (Fig. 3H, I). Be careful: a decrease in caspase 3 level is not a pro-apoptotic marker. What you really show in picture, considering the molecular weight of 32 kDa, is a decrease in pro-caspase-3 level, which is really a pro-apoptotic marker. Pro-caspase-3 mRNA reduction, on the contrary, has a questionable meaning, but not in an apoptotic sense. Furthermore, PARP as an apoptotic marker must be analyzed at the protein level, where the value considered is the ratio between the caspase-cleaved fragment and the full-length form. To evaluate PARP as an apoptotic marker the wb analysis must be repeated considering both the full length and the cleaved form.

Response：Thanks for your questions and explanation. We also found Figure 3C have similar problem.We already have used flow cytometry to detect apoptosis in sh-NC, sh-SNHG17, DOX, sh-SNHG17+dox DoHH2 and Daudi cells, as shown in Figure 3B. Therefore, we have removed Western blot in Figure 3C that was considered unnecessary and incorrect. However, in order to ensure the reliability of the experimental data, we have performed an independent set of experiments. We used the re-analysis the full length parp, including the cleaved form with Western blot. We have included the new experimental results in the supplementary data, as shown in Extended Data Figure S6.

For Fig. 3H, I, The problem was fixed.We used flow cytometry to detect apoptosis rather than measure caspase-3.

Fig 3. DOXO concentration should be indicated at least in the legend 

Response：The concentration of DOX has been indicated in the legend.

318 < 0.01), but no significant effect was observed in SNHG17-MUT cells (Fig. 5FI-II). qPCR results

319 showed that compared with PBMC, the expression of miR-34a-5p in DLBCL cells was significantly

320 downregulated (Fig. 5D, G). In addition, SNHG17 knockdown increased the expression of miR-34a-

321 5p in DoHH2 cells (Fig.5G). This result indicated that miR-34a-32a- 5p DoHH2 is negatively regulated by SNHG17 in this cell line.

Response：We are very sorry that we did not match the data graph with the stated content one by one and made errors in the stated content. We have made corrections and double-checking.

Reply to Editor:Thank you for providing The PLOS ONE style templates. These two documents are very helpful for the revision and improvement of our paper. We have made modifications according to PLOS ONE's style requirements.

 Reply to Editor:Thank you for providing These professional scientific editing service.We have employed the Editage website (www.editage.com) to help us revise our paper,such as language usage, spelling, and grammar. 

Reply to Editor:Yes,we will check Funding Information.

Reply to Editor:OK.

Reply to Editor:Thank you for the reviewer's suggestion. As Reviewer suggested that we have removed the ethical statement written outside the methods section of our manuscript.

6. PLOS ONE now requires that authors provide the original uncropped and unadjusted images underlying all blot or gel results reported in a submission’s figures or Supporting Information files. This policy and the journal’s other requirements for blot/gel reporting and figure preparation are described in detail. 

Reply to Editor: we will resubmit our revised manuscript with uncropped images.

7. Please review your reference list to ensure that it is complete and correct. If you have cited papers that have been retracted, please include the rationale for doing so in the manuscript text, or remove these references and replace them with relevant current references. Any changes to the reference list should be mentioned in the rebuttal letter that accompanies your revised manuscript. If you need to cite a retracted article, indicate the article’s retracted status in the References list and also 

Reply to Editor:Thank you for the reviewer's suggestion. We have reviewed our reference list one by one and have not withdrawn any papers. We ensure that they are complete and correct.

Reviewer's Responses to Questions

Comments to the Author

5. Review Comments to the Author

Reviewer #1: The authors Lu et al., have determined the role of LncRNA SNHG17 in the progression of diffuse large B-cell lymphoma (DLBCL). They have shown that SNHG17 is a ceRNA and acts as a sponge for miR-34a-5p and regulates the EZH2 signaling pathway, and have identified SNHG17 as a potential therapeutic target for DLBCL.

Minor Comments:

The authors should proof read for grammatical and typographical errors.

Line 45, Page 9: miR-34a-5p is typed twice. Please rephrase to convey the interaction between SNHG17 and miR-34a-5p; and miR-34a-5p and EZH2.

Reply to Reviewer # 1:Thank you for pointing out this error for us.It was fixed.

Reviewer #2: This article is interesting and deserves to be published but needs some revisions. In general, English corrections must be made in the drafting of the English form and the organization of the text must be improved, making it more fluid and avoiding repetitions. Legends to figures needs more care. In the introduction section the aim of the paper must be better enucleated. More specific points are in the attached file.

Reply to Reviewer # 2:Thank you for the reviewer's suggestion. We have carefully reviewed and made corrections to our manuscript, and have employed a professional scientific editing service(www.editage.com)recommended by the editor. We have made corrections in language usage and spelling according to the requirements of editors, commentators, and PLOS ONS. In addition, in the introduction section, we have added an explanation of the purpose of the paper.

---

## [Editor Report · Decision Letter 1]

27 Oct 2023

PONE-D-23-18237R1Long non-coding RNA SNHG17 may function as a Competitive Endogenous RNA in diffuse large B-cell lymphoma progression by sponging miR-34a-5pPLOS ONE

Dear Dr. Liao,

Thank you for submitting your manuscript to PLOS ONE. After careful consideration, we feel that it has merit but does not fully meet PLOS ONE’s publication criteria as it currently stands. Therefore, we invite you to submit a revised version of the manuscript that addresses the points raised during the review process.

We look forward to receiving your revised manuscript.

Kind regards,

Lucia Magnelli

Academic Editor

PLOS ONE

Journal Requirements:

Additional Editor Comments:

The Ms by Liao et al. has been greatly improved according to the reviewers' suggestions'. Anyway, the Authors didn't satisfy a requisite of PLOS ONE, as they do not provide original uncropped wb figures. Uncropped images must include the entire blot image with molecular weight standards.

Minor points. Funding section incomplete: include initials of the authors who received each award and the URL of each funder website.

Regarding PARP, although the figure showing both the high molecular weight form and the fragment resulting from the cleavage was inserted in fig s6, results are not modified accordingly (line 301-302)

---

## [Author Response · Author response to Decision Letter 1]

2 Nov 2023

Journal Requirements:

Reply to Editor:Thank you for the reviewer's suggestion.We have checked all the reference,there are not retracted papers in our reference list.We ensure that they are complete and correct.

Additional Editor Comments:

The Ms by Liao et al. has been greatly improved according to the reviewers' suggestions'. Anyway, the Authors didn't satisfy a requisite of PLOS ONE, as they do not provide original uncropped wb figures. Uncropped images must include the entire blot image with molecular weight standards.

Minor points. Funding section incomplete: include initials of the authors who received each award and the URL of each funder website.

Regarding PARP, although the figure showing both the high molecular weight form and the fragment resulting from the cleavage was inserted in fig s6, results are not modified accordingly (line 301-302)

Reply to Editor:Thank you for pointing out this error for us.We uploaded the entire blot images with molecular weight standards(S1_raw_images.pdf).We have fixed Funding section and offered extra-file for funding.We have revised the results about PARP.

---

## [Editor Report · Decision Letter 2]

8 Nov 2023

Long non-coding RNA SNHG17 may function as a Competitive Endogenous RNA in diffuse large B-cell lymphoma progression by sponging miR-34a-5p

PONE-D-23-18237R2

Dear Dr. Liao,

We’re pleased to inform you that your manuscript has been judged scientifically suitable for publication and will be formally accepted for publication once it meets all outstanding technical requirements.

Kind regards,

Lucia Magnelli

Academic Editor

PLOS ONE

Additional Editor Comments (optional):

Dear Dr Liao,

thank you for complying with the requests. Given the new requests from scientific journals, I advise you, for the future, to acquire images of whole wb, with all molecular weights, before cutting the membranes
---

## [Editor Report · Acceptance letter]

11 Nov 2023

PONE-D-23-18237R2 

Long non-coding RNA SNHG17 may function as a competitive endogenous RNA in diffuse large B-cell lymphoma progression by sponging miR-34a-5p 

Dear Dr. Liao:

I'm pleased to inform you that your manuscript has been deemed suitable for publication in PLOS ONE. Congratulations! Your manuscript is now with our production department. 

Kind regards, 

on behalf of

Dr. Lucia Magnelli 

Academic Editor

PLOS ONE